# Targeting CTC Heterogeneity: Aptamer-Based Liquid Biopsy Predicts Outcome in Lung Cancer

**DOI:** 10.3390/cancers17193244

**Published:** 2025-10-06

**Authors:** Alexey V. Krat, Galina S. Zamay, Dmitry V. Veprintsev, Daria A. Kirichenko, Olga S. Kolovskaya, Tatiana N. Zamay, Yury E. Glazyrin, Zoran Minic, Semen A. Sidorov, Valeria A. Komissarova, Ruslan A. Zukov, Maxim V. Berezovski, Anna S. Kichkailo

**Affiliations:** 1Krasnoyarsk Regional Clinical Cancer Center Named After A.I. Kryzhanovsky, Krasnoyarsk 660133, Russia; krat72@inbox.ru (A.V.K.); komissarovava@onkolog24.ru (V.A.K.); zukov_rus@mail.ru (R.A.Z.); 2Laboratory for Biomolecular and Medical Technologies, Prof. V.F. Voino-Yasenetsky Krasnoyarsk State Medical University, Krasnoyarsk 660022, Russia; gzamay@yandex.ru (G.S.Z.); astheno@mail.ru (D.A.K.); olga.kolovskaya@bk.ru (O.S.K.); tzamay@yandex.ru (T.N.Z.); yury.glazyrin@ksc.krasn.ru (Y.E.G.); 3Laboratory for Digital Controlled Drugs and Theranostics, Federal Research Center “Krasnoyarsk Science Center of the Siberian Branch of the Russian Academy of Science”, Krasnoyarsk 660036, Russia; d.v.veprintsev@yandex.ru; 4Department of Chemistry and Biomolecular Sciences, University of Ottawa, Ottawa, ON K1N6N5, Canada; zminic@uottawa.ca

**Keywords:** lung cancer, circulating tumor cells (CTCs), liquid biopsy, DNA aptamers, metastasis, patient survival, diagnostic efficacy, biomarkers

## Abstract

Liquid biopsy, particularly the detection of circulating tumor cells (CTCs), represents a convenient non-invasive approach for cancer diagnosis and monitoring. However, the clinical utility of current methods is limited by their reliance on epithelial markers like EpCAM, which fails to capture all heterogenic blood CTCs due to epithelial–mesenchymal transition. We address this critical limitation by presenting a novel, aptamer-based detection strategy that overcomes the dependency on a single marker. DNA aptamers LC-17 and LC-18, which have high affinity and specificity to different cancer markers, effectively capture tumor cells. We found that these aptamers successfully identified CTCs in lung cancer patients by binding to two specific proteins. The number of CTCs depends on the T-stage and patient survival. Our findings suggest that this aptamer-based liquid biopsy has strong potential for integration into clinical practice.

## 1. Introduction

Lung cancer (LC) incidence varies widely across different countries, ranking second in prevalence worldwide among all oncological diseases. The highest incidence rates of LC are observed in developed countries and are accompanied by high mortality rates [1]. Early detection of this disease contributes to a reduction in mortality among LC patients, as it allows for timely initiation of antitumor therapy. Typically, a tissue biopsy is used to detect malignant tumors and determine their histological type. However, the cost and invasiveness of this procedure limit its application. An alternative to tissue biopsy is liquid biopsy, which allows for the isolation of circulating tumor cells (CTCs) from the patient’s blood that carry all the characteristics of the primary tumor [2]. CTCs represent individual metastatic precursors of tumor formation and have potential value for monitoring the progression of malignant tumors [3]. Elevated CTC levels generally depend on increased tumor mass, low therapy sensitivity, and a poor survival prognosis [4,5].

Numerous studies have confirmed the significant potential of CTCs in clinical practice. The overall diagnostic efficacy of CTCs for detecting LC has shown a sensitivity and specificity [6,7,8,9,10,11,12,13,14]. Despite this, clinical guidelines still do not include their use for diagnosing malignant neoplasms. The main factor limiting the use of CTCs for diagnosis appears to be that circulating tumor cells in the blood are relatively rare compared to other blood cells. Another limiting factor is the high cost of CTC detection, as standard testing typically requires at least three antibodies simultaneously. Furthermore, tumor cells constantly evolve, undergoing epithelial–mesenchymal transitions in the blood, losing some antigens and acquiring others [15], which complicates the search for and identification of CTCs. Therefore, the development of a reliable strategy for detecting CTCs in blood is highly relevant for their use in clinical practice. Currently, various systems are used to search for CTCs, including CellSearch^®^, AdnaTest, ISET^®^, and methods based on quantitative reverse transcription PCR (RT-qPCR) [16,17], among others. For a long time, the only FDA-approved method for detecting circulating tumor cells (CTCs) was the CellSearch^®^ system [18]. Recently, the FDA has also cleared the Parsortix^®^ system for CTC detection in metastatic breast cancer, which utilizes a novel approach for cell separation based on size and deformability [19]. However, the frequency of CTC-positive results obtained using the CellSearch^®^ system is low, primarily due to its reliance on epithelial markers, which do not account for non-epithelial tumor cells. In contrast, the Parsortix^®^ PC1 system can capture and harvest CTCs in a significantly larger proportion of patients, as it enables the identification of CTCs expressing cytokeratin but lacking EpCAM, highlighting the limitations of EpCAM-based approaches [19].

Typically, antibodies are used as recognition elements for CTCs. However, the availability and potential cross-reactivity of antibodies may limit application. Synthetic alternatives to antibodies are aptamers, which act as ligands, binding very tightly to the target molecule [20]. Additionally, aptamers are stable and exhibit low variability during production [21]. Although aptamers have a relatively low molecular weight (approximately ten times less than that of monoclonal antibodies), their complex tertiary structures create a sufficiently large recognition surface area for close interaction with target molecules [22]. Currently, there is no ideal marker capable of reliably and effectively differentiating all CTCs from other blood cells. Aptamers have great potential for detecting CTCs, as they bind to their targets with high affinity and selectivity, and they are more stable than antibodies. Furthermore, aptamers can recognize several non-epithelial biomarkers and can be used across a wide range of experimental conditions and in combination with several functional components.

This study describes a method for detecting CTCs in the blood of LC patients using DNA aptamers that were previously selected against tumor cells derived from the surgically resected lung tumor tissues [23] and investigates their clinical significance, specifically the relationship with primary tumor size, degree of regional lymphatic metastasis, and patient survival.

## 2. Materials and Methods

All experiments involving human tissues were conducted in accordance with the ethical principles set forth in the Declaration of Helsinki. The study protocol was approved by the Local Ethics Committee of the A.I. Kryzhanovsky Krasnoyarsk Regional Clinical Cancer Center (Protocol No. 8, 16 March 2011, Krasnoyarsk, Russia) and Prof. V.F. Voino-Yasenetsky Krasnoyarsk State Medical University (Protocol No. 37, 31 January 2012). Informed consent was obtained from all participants prior to their inclusion in the study.

### 2.1. Identification of Aptamer Target Proteins in Circulating Tumor Cells

To determine the target proteins of the LC-17 aptamer in CTCs, blood samples from seven LC patients were used. The CTC samples derived from each patient were incubated either with aptamers or nonspecific control sequence separately. Additional controls included one LC cell culture preparation and whole blood samples from three healthy volunteers. For protein extraction, magnetic particles coated with gold and functionalized with thiolated LC-17 aptamer (5′-/5ThioMC6-D/CTC CTC TGA CTG TAA CCA CGC TTT TGT CTT TAG CCG AAT TTT ACT AAG CCG GGC TGA TCA GCA TAG GTA GTC CAG AAG CC-3′)-were employed (Lumiprobe, Moscow, Russia). As a negative control, a hybrid of a thiolated primer and a nonspecific DNA sequence consisting of AG repeats was used (5′-/5ThioMC6-D/CTC CTC TGA CTG TAA CCA CGA GAG AGA GAG AGA GAG AGA GAG AGA GAG AGA GAG AGA GAG-3′) (Lumiprobe, Moscow, Russia). All experiments were performed in triplicate.

The CTCs for aptamer target identifications were isolated from whole blood as described in Section 2.1. The isolated CTCs as well as control culture cells were washed, lysed by 0.1% n-dodecyl β-D-maltoside (DDM) (Thermo Scientific, Waltham, MA, USA) for 30 min, and centrifuged at 15,000× *g*. The protein-containing supernatant was collected and incubated with masking yeast RNA. The gold-coated magnetic particles were first conjugated with thiolated aptamers (or the AG hybrid) for 16 h and then added to the protein solution. Following 30 min of incubation, the particles were immobilized using a magnet and washed five times using the phosphate buffer. The target proteins were then eluted from the particles after 30 min of incubation in 8 M urea and collected as a solution after magnetic separation.

Target protein concentrations were measured by NanoDrop spectrophotometer (Thermo Scientific, Waltham, MA, USA). Equivalent to 2 μg volumes of protein solutions were reduced with dithiothreitol, alkylated with iodoacetamide, and digested with trypsin for 16 h at 37 °C according to the manufacturer’s protocol (Thermo Scientific, Waltham, MA, USA). The derived peptide samples were desalted using C18 pipette tips (Thermo Scientific, Waltham, MA, USA). The dried samples were redissolved in 0.1% formic acid in water (phase “A”) and injected into the LC system for analysis.

Liquid chromatography–mass spectrometry was conducted using the Easy-nLC 1000 HPLC system and the Orbitrap Velos Pro high-resolution mass spectrometer (Thermo Scientific, Waltham, MA, USA). An Easy-spray C18 chromatography column (15 cm length, 75 µm inner diameter, 3 µm particle size) combined with the ion emitter was used for peptide separation and electrospray injection. An analytical gradient from 0% to 40% of phase “B” (0.1% formic acid in acetonitrile) was run for 40 min at 300 nL/min flow, followed by a column cleaning and regeneration stage with an increase in the “B” content to 80% in 15 min, holding at 80% of “B” for 5 min and returning to 0% of “B” in 5 min. A data-dependent mass spectrometry acquisition was performed in tandem regime with the switching between a primary peptide scan in high-resolution mode and 10 secondary scans of ions with the highest intensities fragmented by collision-induced dissociation (CID). The primary ion mass scanning was performed by the Orbitrap mass analyzer with a resolution of 60,000. The secondary fragment scanning was performed by the ion trap with the standard resolution. After each primary and top 10 scanning cycle, the previously fragmented peptides were placed in a temporary exclusion list for 60 s, and the next 10 most intense ions were selected from the primary spectrum for fragment mass scanning.

The tandem spectra files obtained from the whole set of samples were processed simultaneously using the MaxQuant freeware package (2.0 version). A protein search was performed via the current SwissProt database with human protein restriction at a false discovery rate (FDR) of 0.1. The presence of distinct proteins in a larger number of samples of the experimental set compared to the control sample set was used as a rationale for selecting target proteins. No aptamer-specific proteins were detected in the equally prepared blood samples derived from healthy volunteers.

### 2.2. Isolation of Circulating Tumor Cells

The study was approved by the Local Ethics Committee of the A.I. Kryzhanovsky Krasnoyarsk Regional Clinical Oncology Dispensary (Protocols No. 8/2016, 16 March 2016, and No. 55, 28 December 2022).

Peripheral blood samples (4 mL) from patients were centrifuged at 430× *g* for 10 min, after which the plasma was removed.

To prevent CTC adhesion to plastic, pipette tips were pre-rinsed with 30% BSA before further processing. The pellet (1.5 mL) was transferred into 15 mL centrifuge tubes, and 10 mL of 0.42% NH_4_Cl with heparin was added. Samples were incubated for 10 min on a shaker to lyse erythrocytes, followed by centrifugation at 2300× *g* for 5 min. The supernatant was discarded, and the pellet was resuspended in 3 mL of 0.2% NaCl with heparin for partial lymphocyte lysis, pipetted, and incubated on a shaker for 60 min.

After lysis, cells were centrifuged at 2300× *g* for 5 min. The supernatant was removed, and 3 mL of 0.9% NaCl with heparin was added to restore cell morphology. The suspension was gently pipetted and left for 10 min, then centrifuged again at 2300× *g* for 5 min. The supernatant was discarded, and the pellet was resuspended in 1 mL of 0.9% NaCl. The sample was filtered through a 70-µm cell strainer into a new 15 mL tube, followed by an additional wash with 1 mL of 0.9% NaCl with heparin, resuspension, and a second filtration step.

Finally, the samples were centrifuged, the supernatant was removed, and the cells were stained with FAM-labeled aptamers for 30 min, transferred onto a glass slide, air-dried, and fixed with 70% methanol for 5–10 min in the dark. The cells were then stained using the Romanowsky–Giemsa method.

Samples were analyzed using an Olympus BX51 fluorescence microscope (Olympus Corporation, Tokyo, Japan).

Aptamers LC-17 and LC-18 were early obtained by Zamay G.S. et al. [24]. The aptamer’s sequences are presented in Table 1.

## 3. Results

### 3.1. Identification of CTC-Specific Target Proteins for Aptamer LC-17

To identify target proteins of aptamer LC-17 in CTCs, blood samples from 7 NSCLC patients were used. Two proteins were detected as reliably predominant in aptamer-derived samples compared to controls and were not detected in blood samples of healthy individuals. The most abundant protein was neutrophil defensin 1 (DEFA1). Peroxiredoxin-2 (PRDX2) also had elevated LFQ values in aptamer LC-17 samples. The only protein that was most abundant in both tissue and CTC samples was neutrophil defensin.

Table 2 lists LC-17 aptamer targets isolated from CTCs, LC cell cultures, and healthy donor blood using gold-coated magnetic particles, with mass-spectrometric verification.

### 3.2. Detection of CTCs in Blood Samples from NSCLC Patients After Blood Cell Lysis

CTCs, isolated from blood via filtration subsequent to erythrocyte and lymphocyte lysis, were identified by staining with FAM-labeled LC-17 and LC-18 aptamers and subsequent analysis using fluorescence and light microscopy (Figure 1).

A comprehensive study quantifying circulating tumor cells (CTCs), circulating tumor microemboli (CTM), and apoptotic cells (ATs) using fluorescence microscopy—following erythrocyte and lymphocyte lysis—included 43 LC patients (data shown in Table 3).

Our analysis of aptamer-positive CTC counts in blood samples from LC patients revealed significant associations with both primary tumor size (T stage) and lymph node involvement (N stage). Notably, we identified a correlation between CTC numbers and overall survival in NSCLC patients.

Using the Mann–Whitney U test, we compared CTC counts across different T stages (T1–T4). As median CTC counts were similar in T1–T3 groups (*p* > 0.05), these were combined for analysis. A statistically significant difference (*p* = 0.012) was observed between the combined T1–3 group and T4 tumors, demonstrating CTC count dependence on primary tumor size (Figure 2).

Our findings demonstrate that locally advanced tumors (T4; >7 cm) with extensive invasion of adjacent structures (diaphragm, mediastinum, great vessels, etc.) or metastatic ipsilateral lobe nodes show higher aptamer-positive CTC counts in peripheral blood (Figure 2).

No significant difference was observed between N0 and N1 (metastasis to ipsilateral peribronchial/hilar nodes). Significant CTC count elevation was found when comparing N0-1 vs. N2 (Figure 3). These results establish correlations between CTC abundance and degree of mediastinal lymphatic invasion (N stage).

Building upon the demonstrated correlation between CTC counts and disease progression in NSCLC, we further investigated the clinical prognostic value of aptamer-positive CTCs by analyzing overall survival across patient groups stratified by CTC burden (0–2 vs. >2 cells/4 mL blood). As shown in Figure 4, this clinically relevant threshold revealed significant survival differences (log-rank test, *p* = 0.044), with patients exhibiting elevated CTC counts (>2 cells/4 mL) demonstrating poorer outcomes. Patients with low CTC counts (0–2 cells/4 mL blood) demonstrated longer median overall survival (33 months; 95% CI: 16–93) compared to those with elevated CTC counts (>2 cells/4 mL; 15 months; 95% CI: 5–59).

Thus, it has been demonstrated that using our aptamers, the CTC count in the peripheral blood of LC patients serves as a significant prognostic and clinically relevant parameter. It exhibits statistical dependencies with the primary tumor spread, regional metastasis level, and overall survival rates and promises the potential utility for risk stratification and survival prediction.

## 4. Discussion

Despite significant interest in liquid biopsy as a tool for the diagnosis and monitoring of oncological diseases, it has not yet become a standard analytical material and is primarily used as an additional test.

Recently, new technologies have emerged in science that offer the potential to solve existing problems and develop effective means of diagnosis and therapy based on highly specific artificial antibodies (aptamers) to any biological targets. Aptamers selected for key proteins involved in oncogenic transformation are strong candidates for the detection of circulating tumor cells (CTCs) in the blood of patients.

One of the key challenges in working with aptamers is the selection of targets and the generation of DNA oligonucleotides that will specifically bind to epitopes of cancer biomarkers. Despite a large number of studies on this topic, which have been steadily increasing since 1990, the number of aptamer-based drugs that have passed preclinical trials and reached the market is very small.

Aptamers selected against primary materials have a higher specificity for binding to cancer markers in real clinical samples compared to aptamers selected against recombinant proteins; however, the problem of accurate target identification remains.

The protein targets of aptamers LC-17 and LC-18 were previously identified and validated from the binding to primary lung tumors according to the study by G. Zamay et al. [24] (Table 4).

Vimentin, a type III intermediate filament protein in mesenchymal cells, has been the subject of intensive research worldwide in recent years due to its complex biological functions and critical role in the epithelial–mesenchymal transition during tumor progression. Additionally, vimentin [25] modulates tumor cell migration, invasion, and adhesion [26]. Vimentin is highly expressed in metastatic cancer, and its expression depends on poor patient prognosis [27]. Thus, vimentin isolated from lung tumor tissue is a well-established cancer biomarker.

Lamin A proteins are fundamental components of the nuclear lamina. Altered lamin A expression depends on malignant transformation in certain cancers [25].

Defensins are a family of small cationic peptides containing six cysteine residues linked by three intramolecular disulfide bonds, with a central β-sheet dominating their structure. These proteins can induce DNA damage and trigger tumor cell apoptosis. In the tumor microenvironment, defensins act as chemoattractants for immune cell subsets, such as T cells, immature dendritic cells, monocytes, and mast cells. Moreover, by activating target leukocytes, defensins generate pro-inflammatory signals. Overall, defensins play a crucial role in tumor immunity [28].

Alpha/beta-tubulin subunits are among the key oncogenes contributing to malignant progression. Previous studies have shown that dysregulation of their structural modifications promotes carcinogenesis [29].

In this study, the targets of the LC-17 aptamer for circulating tumor cells were identified. These are two proteins: neutrophil defensin and peroxiredoxin-2.

Peroxiredoxins are antioxidant enzymes that protect cells from oxidative stress and reduce the intracellular accumulation of reactive oxygen species. Their involvement has been demonstrated in immune response, regulation of cell growth, apoptosis, differentiation, and metabolism. Studies have shown that peroxiredoxin expression is associated with poor prognosis in various cancers [30]. Research indicates that peroxiredoxin overexpression strongly correlates with CD133+ CD44+ cells in colon cancer tissues. Its inhibition suppresses tumor maintenance, migration, and invasion, while also reducing the risk of liver metastasis. Peroxiredoxin is suggested as a potential therapeutic target for malignant neoplasms [31].

Neither neutrophil defensin nor peroxiredoxin-2 was isolated from healthy donors’ blood. Since hemoglobin was present in all sample types, it can be considered a contaminant.

The enumeration of CTCs and statistical analysis of their levels in patient blood revealed a correlation between CTC count and both the stage of the tumor process and the extent of regional metastasis in NSCLC patients. Higher counts of aptamer-positive targets were seen in stage T4 NSCLC patients compared to stages T1 through T3. Patients with mediastinal lymph node involvement (N2 category) exhibited elevated aptamer-positive target levels versus non-metastatic LC (N0); patients with intrapulmonary or peribronchial lymph node metastases (N1) also showed increased levels. The presence of CTCs in blood was linked to significantly worse overall survival in NSCLC.

This indicates that the described method could potentially be used for disease monitoring in LC patients, offering a safer alternative to computed tomography (CT) and positron emission tomography/computed tomography (PET/CT) that utilizes radionuclides.

Thus, the technique for detecting CTCs in blood using aptamers has potential for clinical application.

The measurement of CTC levels using a DNA aptamer-based method is proposed as a valuable tool for clinical management, including staging (assessment of tumor dissemination), monitoring disease progression, and evaluating response to treatment.

Based on these findings, a scheme for monitoring LC patients using the aptamer-based CTC detection method has been developed (Figure 5).

The proposed aptamer-based liquid biopsy scheme for LC monitoring necessitates more rigorous evaluation. The primary obstacle in CTC detection is the low frequency of these events in peripheral blood. Moreover, sample loss is unavoidable throughout the entire workflow, from blood collection to preparation, and may also occur during the CTC counting process itself. Consequently, interpretation of the results should be based on relative CTC levels rather than on absolute counts. Defining these reference levels will only be possible after accumulating data from a large-scale study involving cohorts of patients with LC, benign pulmonary conditions, other malignancies, and healthy donors. Additionally, CTC levels must be assessed in LC patients undergoing therapy and during their clinical follow-up.

Therefore, the certification process for clinical translation of these techniques can only commence after establishing reference CTC levels for the various groups, including treated LC patients.

Another crucial consideration for the preclinical and clinical advancement of aptamer-based systems is a comprehensive understanding of the target, particularly the specific epitope to which the aptamer binds. This necessitates a thorough investigation of the aptamer’s conformation and its potential cross-reactivity with proteins of similar structure. Methods such as small-angle X-ray scattering (SAXS), circular dichroism (CD) spectroscopy, and nuclear magnetic resonance (NMR) spectroscopy are essential for determining the aptamer’s three-dimensional structure. The authors plan to pursue these investigations to advance the development of novel, affordable, and minimally invasive methods for LC diagnosis and monitoring.

## 5. Conclusions

This study demonstrates the clinical potential of an aptamer-based method for detecting circulating tumor cells (CTCs) in patients with non-small cell lung cancer (NSCLC). We established that DNA aptamers LC-17 and LC-18 effectively identify CTCs in the peripheral blood of NSCLC patients using microscopy. A key advantage of this approach over EpCAM-dependent platforms is its ability to capture a broad spectrum of CTC phenotypes. This multi-target aptamer recognition strategy is independent of cell size, physical properties, and epithelial markers, making it a significant advance for liquid biopsy applications in advanced NSCLC, where tumor heterogeneity is a significant challenge. Thus, it offers a promising, non-invasive tool for monitoring disease progression, assessing treatment efficacy, and predicting patient outcomes. We therefore propose the integration of this aptamer-based CTC assay into standard clinical practice for NSCLC patient management. To fully realize this potential in clinics, future work must focus on validating the method in larger, multi-center cohorts and exploring its utility in guiding personalized treatment decisions. Furthermore, the adaptability of the aptamer-based recognition principle suggests its potential applicability could be extended to other cancer types where CTC heterogeneity impedes conventional detection methods.

## Figures and Tables

**Figure 1 cancers-17-03244-f001:**
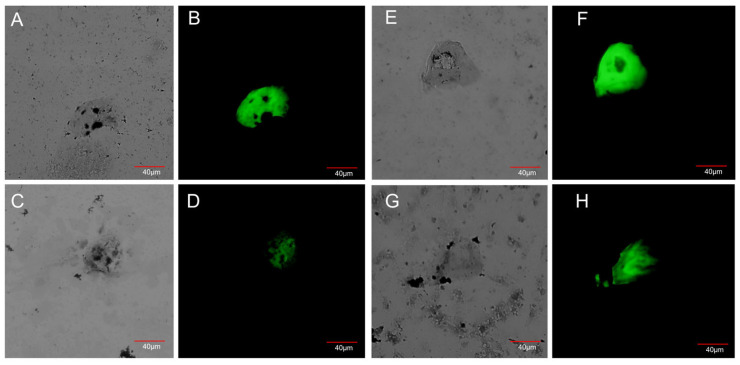
Blood smears from blood of NSCLC patients. CTCs were stained by FAM-labeled LC-17 and LC-18 (labeled together) and Romanovsky–Giemsa dye. (**A**,**C**,**E**,**G**)—light microscopy, (**B**,**D**,**F**,**H**)—fluorescent microscopy.

**Figure 2 cancers-17-03244-f002:**
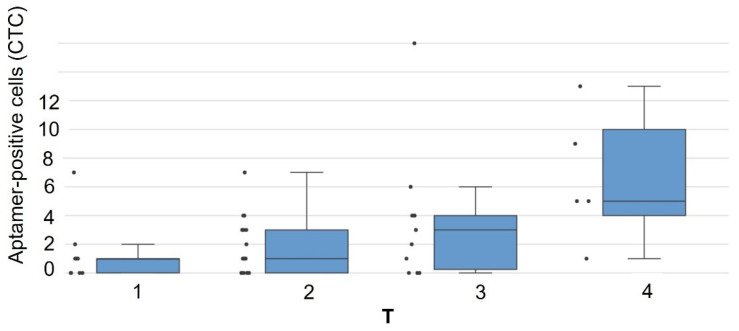
Association between aptamer-positive CTC count in 4 mL of blood and primary tumor size (T stage) in NSCLC patients. The box plot shows significantly higher CTC counts in T4 tumors compared to combined T1–T3 stages (Mann–Whitney U test, *p* = 0.012).

**Figure 3 cancers-17-03244-f003:**
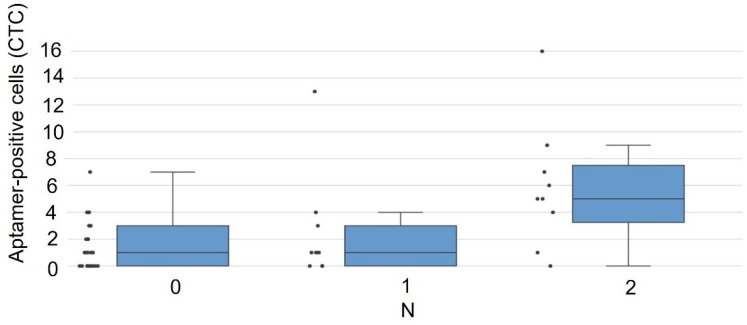
Correlation between aptamer-positive CTC counts in 4 mL of blood and regional lymph node metastasis (N stage) in NSCLC patients. Box plot demonstrates significantly higher CTC counts in patients with N2 mediastinal nodal involvement compared to combined N0–N1 groups (Mann–Whitney U test, *p* = 0.014).

**Figure 4 cancers-17-03244-f004:**
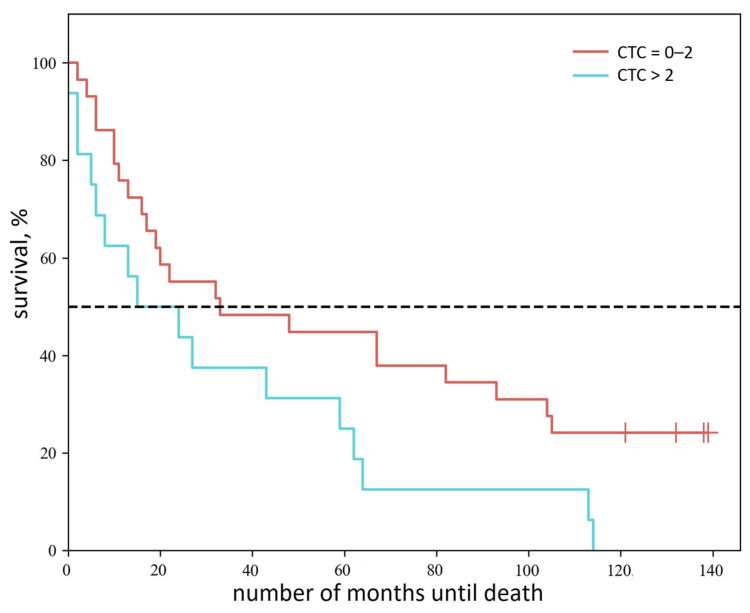
Survival analysis of NSCLC patients stratified by aptamer-positive CTC count. The curve demonstrates significantly worse overall survival in patients with >2 CTCs/4 mL blood (blue) compared to those with 0–2 CTCs/4 mL (red).

**Figure 5 cancers-17-03244-f005:**
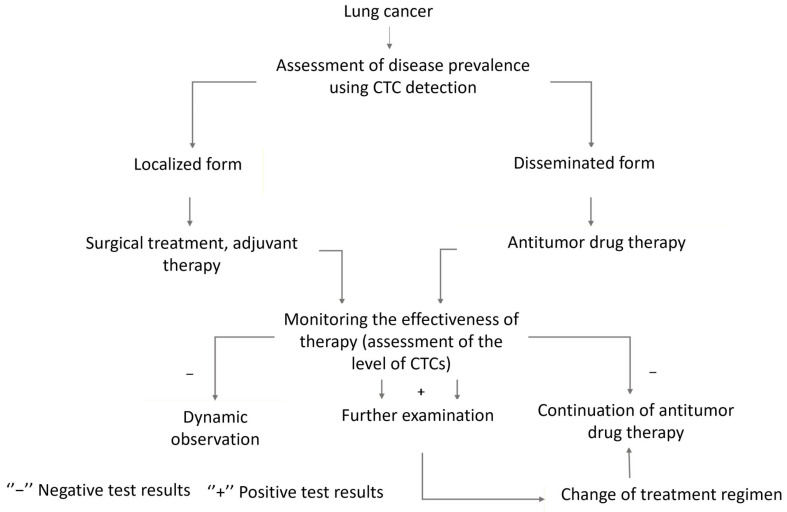
Optimizing a lung cancer patient management strategy using aptamer-based diagnostic tools.

**Table 1 cancers-17-03244-t001:** Aptamers LC-17 and LC-18 sequences [24].

Aptamer Name	Aptamer Sequence
LC-17	5′-CTC CTC TGA CTG TAA CCA CGC TTT TGT CTT TAG CCG AAT TTT ACT AAG CCG GGC TGA TCA GCA TAG GTA GTC CAG AAG CC-3′
LC-18	5′-CTC CTC TGA CTG TAA CCA CGT GCC CGA ACG CGA GTT GAG TTC CGA GAG CTC CGA CTT CTT GCA TAG GTA GTC CAG AAG CC-3′

**Table 2 cancers-17-03244-t002:** Sample types used for LC-17 aptamer target isolation and reliably identified proteins.

Target Protein	CTCs	Lung Cancer Cell Culture	Healthy Donor Blood
Neutrophil defensin 1	Present	Present	Absent
Peroxiredoxin-2	Present	Absent	Absent

**Table 3 cancers-17-03244-t003:** Number of CTCs in the peripheral blood of lung cancer patients and patient survival duration.

№	Histological Type of LC	Stage	TNM	Patient Survival Duration, Months	Number of CTCs, Units in 4 mL of Blood
1	Squamous cell lung cancer	IB	T2N0M0	105	0
2	Pulmonary adenocarcinoma	IV	T3N1M1	10	0
3	Pulmonary adenocarcinoma	IA	T1N0M0	93	0
4	Squamous cell lung cancer	IIB	T3N0M0	17	0
5	Squamous cell lung cancer	IIIB	T3N2M0	2	0
6	Squamous cell lung cancer	IIB	T2N1M0	67	0
7	Pulmonary adenocarcinoma	IIA	T2N0M0	6	0
8	Squamous cell lung cancer	IIB	T2N1M0	33	0
9	Pulmonary adenocarcinoma	IIA	T2N0M0	Alive	0
10	Pulmonary adenocarcinoma	IA	T1N0M0	Alive	0
11	Pulmonary adenocarcinoma	IIA	T2N0M0	16	0
12	Pulmonary adenocarcinoma	IIIB	T4N2M0	82	0
13	Squamous cell lung cancer	IA	T1N0M0	20	1
14	Small cell lung cancer	IV	T2N0M1	19	1
15	Pulmonary adenocarcinoma	IV	T2N2M1	Alive	1
16	Pulmonary adenocarcinoma	IB	T2N0M0	105	1
17	Pulmonary adenocarcinoma	IIA	T2N0M0	22	1
18	Pulmonary adenocarcinoma	IIA	T2N0M0	32	1
19	Adenosquamous lung carcinoma	IIA	T2N0M0	48	6
20	Pulmonary adenocarcinoma	IIA	T2N0M0	6	1
21	Squamous cell lung cancer	IA	T1N0M0	11	1
22	Pulmonary adenocarcinoma	IA	T1N0M0	Alive	1
23	Pulmonary adenocarcinoma	IIA	T2N0M0	Alive	1
24	Pulmonary adenocarcinoma	IIIA	T2N2M0	67	1
25	Pulmonary adenocarcinoma	IIA	T2N0M0	4	1
26	Pulmonary adenocarcinoma	IA	T1N0M0	Alive	2
27	Squamous cell lung cancer	III A	T3N0M0	133	2
28	Squamous cell lung cancer	III A	T3N0M0	10	2
29	Pulmonary adenocarcinoma	IIB	T2N2M0	13	3
30	Squamous cell lung cancer	IIIB	T3N2M0	9	3
31	Pulmonary adenocarcinoma	IIA	T2N0M0	1	3
32	Squamous cell lung cancer	IIIB	T3N2M0	113	3
33	Squamous cell lung cancer	IIB	T2N1M0	5	4
34	Squamous cell lung cancer	IIA	T2N0M0	27	4
35	Squamous cell lung cancer	IIIA	T3N0M0	59	4
36	Pulmonary adenocarcinoma	IB	T2N0M0	43	4
37	Pulmonary adenocarcinoma	IIIB	T4N2M0	113	4
38	Squamous cell lung cancer	IV	T4N2M1	3	5
39	Squamous cell lung cancer	IIIA	T2N2M0	6	5
40	Pulmonary adenocarcinoma	IA	T1N0M0	2	6
41	Pulmonary adenocarcinoma	IIIA	T2N2M0	62	7
42	Squamous cell lung cancer	IIIB	T4N2M0	64	7
43	Squamous cell lung cancer	IIIB	T4N1M0	15	9

**Table 4 cancers-17-03244-t004:** Protein targets of aptamers LC-17 and LC-18 [24].

LC-17	LC-18
Neutrophil defensin	Vimentin
Tubulin	Lamin A/C

## Data Availability

The data that support the findings of this study are available from the corresponding author upon reasonable request.

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
