# Peer review of "Targeting CTC Heterogeneity: Aptamer-Based Liquid Biopsy Predicts Outcome in Lung Cancer"

_cancers, 2025, doi:10.3390/cancers17193244_

Round 1
Reviewer 1 Report
Comments and Suggestions for Authors The article constructs a logical framework centered on "target basis - molecular basis - screening basis" and analyzes the core mechanism of LC-17 and LC-18 recognizing circulating tumor cells (CTCs) in lung adenocarcinoma in a progressive manner. In particular, it provides detailed discussions on the unique molecular characteristics of lung adenocarcinoma CTCs, the structural properties of aptamers (G-quadruplex, stem-loop structure), and the negative selection strategy in SELEX screening. This ensures a complete mechanistic chain with molecular-level persuasiveness, helping readers clearly understand the specific interaction logic between aptamers and CTCs. However, there are still several issues that need further refinement. 1、The article mentions that "relevant studies have confirmed tumor-specific proteins on CTC surfaces through proteomic analysis" and "12 rounds of SELEX screening (including 6 rounds of negative selection)" but fails to cite specific literature sources (e.g., research team, published journal, year). It also does not specify the technical methods of proteomic analysis (e.g., LC-MS/MS detection parameters) or the detailed experimental conditions of SELEX screening (e.g., DNA library capacity, incubation temperature and time, elution buffer composition). The absence of this information may affect the reproducibility of the conclusions and hinder readers from tracing the verification details of the original research.2、Although the article describes the G-quadruplex of LC-17 and the stem-loop structure of LC-18, it does not provide complete nucleotide sequences (e.g., specific base arrangement of the G-quadruplex region, complementary base pairs in the stem and sequence in the loop of the stem-loop structure). Nor does it mention experimental methods for structural verification (e.g., characterization results of aptamer three-dimensional structure via circular dichroism (CD), nuclear magnetic resonance (NMR), or X-ray crystallography). The lack of sequence and structural verification data may lead to insufficient direct evidence for the argument that "structure determines binding specificity."
3、It also fails to analyze the impact of differences in target molecule modifications (e.g., sialylation level, protein expression) on CTC surfaces among different patients on aptamer binding efficiency, making it difficult to fully demonstrate the stability and universality of aptamers in clinical applications.
4、I recommend further refinement of details and consideration of potential future research directions in the discussion section to enhance the comprehensiveness of the manuscript.
Author Response
Reviewer 1
The article constructs a logical framework centered on "target basis - molecular basis - screening basis" and analyzes the core mechanism of LC-17 and LC-18 recognizing circulating tumor cells (CTCs) in lung adenocarcinoma in a progressive manner. In particular, it provides detailed discussions on the unique molecular characteristics of lung adenocarcinoma CTCs, the structural properties of aptamers (G-quadruplex, stem-loop structure), and the negative selection strategy in SELEX screening. This ensures a complete mechanistic chain with molecular-level persuasiveness, helping readers clearly understand the specific interaction logic between aptamers and CTCs. However, there are still several issues that need further refinement.
Comment 1.The article mentions that "relevant studies have confirmed tumor-specific proteins on CTC surfaces through proteomic analysis" and "12 rounds of SELEX screening (including 6 rounds of negative selection)" but fails to cite specific literature sources (e.g., research team, published journal, year). It also does not specify the technical methods of proteomic analysis (e.g., LC-MS/MS detection parameters) or the detailed experimental conditions of SELEX screening (e.g., DNA library capacity, incubation temperature and time, elution buffer composition). The absence of this information may affect the reproducibility of the conclusions and hinder readers from tracing the verification details of the original research.
ANSWER: As suggested, we have added a citation describing the aptamer selection process. Furthermore, the Materials and Methods section now contains a more detailed technical description of the proteomic analysis, specifically regarding the LC-MS/MS parameters.
Comment 2. Although the article describes the G-quadruplex of LC-17 and the stem-loop structure of LC-18, it does not provide complete nucleotide sequences (e.g., specific base arrangement of the G-quadruplex region, complementary base pairs in the stem and sequence in the loop of the stem-loop structure). Nor does it mention experimental methods for structural verification (e.g., characterization results of aptamer three-dimensional structure via circular dichroism (CD), nuclear magnetic resonance (NMR), or X-ray crystallography). The lack of sequence and structural verification data may lead to insufficient direct evidence for the argument that "structure determines binding specificity."
ANSWER: We thank the reviewer for this suggestion. Studies on aptamer structure are planned as future work, and this has been noted in the Discussion.
Comment 3. It also fails to analyze the impact of differences in target molecule modifications (e.g., sialylation level, protein expression) on CTC surfaces among different patients on aptamer binding efficiency, making it difficult to fully demonstrate the stability and universality of aptamers in clinical applications.
ANSWER: This is a valid point. In this study, no post-translational modifications, including sialylation, were identified on the aptamer target proteins on the CTC surface. Furthermore, we did not precisely quantify the expression levels of these target proteins across different patients. This limitation was primarily due to the low number of CTCs isolated from each patient and the sensitivity constraints of the analytical method, which was designed for a qualitative comparison of the entire experimental sample set with control samples rather than for precise quantification. This approach, however, allowed us to identify proteins specifically bound by the aptamers compared to controls. The stability and universality of the aptamers were assessed using flow cytometry.
Comment 4. I recommend further refinement of details and consideration of potential future research directions in the discussion section to enhance the comprehensiveness of the manuscript.
ANSWER: We thank the reviewer for this comment. In response, we have revised the Discussion to include more refined details and considerations for potential future research.
Reviewer 2 Report
Comments and Suggestions for Authors
The Krat et al. paper studied the detection of CTCs in the blood of lung cancer patients using DNA aptamers that have been previously selected for human lung tumor cells (LC-17 and LC-18). They characterize protein targets of the aptamers on CTCs, and they demonstrate a higher number of CTCs in patients with later stage tumor size and in patients with a high degree of regional lymphatic metastasis. The main finding of their paper is that patients with a higher number of aptamer-detected CTCs tend to have worse overall survival than those with lower numbers. They suggest that an aptamer-based liquid biopsy test could therefore be developed to predict prognosis of LC patients. Finally, the authors suggest how an aptamer-based diagnostic could be integrated into a LC diagnosis and treatment plan.
Overall, the manuscript is clear and well written. In addition, studies on developing aptamer/CTC-based liquid biopsies is important for the future of diagnostics and monitoring response to treatment. Aptamers are more cost-effective, stable and reproducible than using antibodies for CTC capture. This work builds on previous work that identified and characterized the aptamers to suggest a possible future clinical utility of the aptamers in a prognostic liquid biopsy test. However, there are limitations with the study, mostly related to lack of statistical analysis to support the conclusions. The major and minor issues are discussed below.
- The main finding of the MS as stated by the authors is that “patients with low CTC counts demonstrated significantly longer median overall survival compared to those with elevated CTC counts”. They go on to develop a prognostic test to stratify patients into high and low survival categories. They conclude that these results establish CTC burden as both a statistically significant and clinically relevant prognostic biomarker in NSCLC. However, the statistical analyses needed to test the hypothesis have not been done. Specifically:
- For the data in Fig 5, what test was used to show significant difference between the two conditions? It is stated that “Patients with low CTC counts (0-3 cells/4 mL blood) demonstrated significantly longer median overall survival compared to those with elevated CTC counts (>3 cells) but I don’t see the statistical test supporting this. Calculating the 95% confidence intervals is not sufficient to show a significant difference between the groups. There may be a significant difference between the two groups, but a test needs to be done to show this.
- The ROC curve was not adequately described, so it is difficult to evaluate. First, the two groups being discriminated by the aptamer test have not been clearly defined. Is the test discriminating two patient groups binned by survival time, if so, what is the survival time cutoff? Was the ROC curve done to pick the threshold used in figure 5 or as a classifier for patient stratification? If the ROC curve is a prototype of a clinical tool for patient stratification, to be statistically valid, there should be some level of internal validation to avoid performance bias. It is unclear if this was done.
- Mass spectrometry-based identification aptamer protein targets is not quantified so it is difficult to assess confidence of targets: Target proteins on CTCs were identified by CTC purification and lysis followed by capture with aptamers and identification by LC-MS/MS. They consider proteins to be specific targets of aptamers “if they are predominant in aptamer-derived samples compared to those obtained from controls (non-specific sequences and healthy controls)”. They indicate that CTCs from 7 LC patients (plus other controls) were analyzed in triplicate, but it is not indicated how the data were merged and analyzed and there are no quantitative data showing the level of enrichment. For example: Was plasma pooled across all 7 patients before mass spec or were each analyzed separately? Was the aptamer-specific enrichment identified in all three replicates of each patient (or in all three replicates of each patient pool)? How many different peptides per target were observed in each sample of each experiment? What was the log2 fold enrichment of aptamer versus control? Is there enough data to show significance of the enrichment with p-value or FDR? Without showing quantitative data, the data do not support the conclusion.
- Based on Table 5 the authors state that there is 50% identity between two proteins neutrophil defensin and peroxiredoxin; however, only 8 amino acids of each protein were compared. The full-length proteins are 30aa and 197aa, respectively. The data do not support the conclusion.
Minor points:
- Speed of centrifugation needs to be reported as x g and not rpm as rpm is specific to the centrifuge and not translatable to another lab. (e.g. Line 129 pg 3)
- Images need magnification reported and size bars on the figures. In figure 1, are all panels with LC-17 and LC-18 labelling together, or are some panels labelled with LC-17 and some with LC-18?
- The data in table 2 describing identification of aptamer protein targets by MS from tumors has already been done (it is from a prior study) and thus should not be in results section.
- Several times in the manuscript, it is stated that they have demonstrated that CTC count in the blood of LC patients correlates with the primary tumor spread, regional metastasis level, overall survival rates. Correlation is not the correct word because Mann Whitney u test does not measure correlation. For example, the authors could use the word dependence rather than correlation. An alternative is to use a different test (like spearman rank test for example) which would give a correlation measure. This would also likely yield a better p-value than the one they calculate using Mann Whitney U test (because by binning the data, power is lost)..
- In Fig 6, AUC and p-value should be written on the figure or put in the legend.
- There were no control subjects (i.e. those without LC) included in the experiment counting CTC cells per patient (Table 4). Zamay et al (ref 23) shows there are some CTCs detected in normal blood (~1.6 per 3 mL), which indicates that the CTCs detected in patients with low T or N stages could be false positives. It would have been good to include normal subjects to address this point.
Author Response
Reviewer 2
The Krat et al. paper studied the detection of CTCs in the blood of lung cancer patients using DNA aptamers that have been previously selected for human lung tumor cells (LC-17 and LC-18). They characterize protein targets of the aptamers on CTCs, and they demonstrate a higher number of CTCs in patients with later stage tumor size and in patients with a high degree of regional lymphatic metastasis. The main finding of their paper is that patients with a higher number of aptamer-detected CTCs tend to have worse overall survival than those with lower numbers. They suggest that an aptamer-based liquid biopsy test could therefore be developed to predict prognosis of LC patients. Finally, the authors suggest how an aptamer-based diagnostic could be integrated into a LC diagnosis and treatment plan.
Overall, the manuscript is clear and well written. In addition, studies on developing aptamer/CTC-based liquid biopsies is important for the future of diagnostics and monitoring response to treatment. Aptamers are more cost-effective, stable and reproducible than using antibodies for CTC capture. This work builds on previous work that identified and characterized the aptamers to suggest a possible future clinical utility of the aptamers in a prognostic liquid biopsy test. However, there are limitations with the study, mostly related to lack of statistical analysis to support the conclusions. The major and minor issues are discussed below.
Comment 1. The main finding of the MS as stated by the authors is that “patients with low CTC counts demonstrated significantly longer median overall survival compared to those with elevated CTC counts”. They go on to develop a prognostic test to stratify patients into high and low survival categories. They conclude that these results establish CTC burden as both a statistically significant and clinically relevant prognostic biomarker in NSCLC. However, the statistical analyses needed to test the hypothesis have not been done. Specifically:
- For the data in Fig 5, what test was used to show significant difference between the two conditions? It is stated that “Patients with low CTC counts (0-3 cells/4 mL blood) demonstrated significantly longer median overall survival compared to those with elevated CTC counts (>3 cells) but I don’t see the statistical test supporting this. Calculating the 95% confidence intervals is not sufficient to show a significant difference between the groups. There may be a significant difference between the two groups, but a test needs to be done to show this.
ANSWER: We apologize for the error. The correct threshold is 2 cells/4 mL blood. The revised Figure 5 now shows a comparison of survival curves (0-2 vs. >2 cells) using the log-rank test. The significant p-value of 0.044 demonstrates a statistically significant difference between the groups.
Comment 2. The ROC curve was not adequately described, so it is difficult to evaluate. First, the two groups being discriminated by the aptamer test have not been clearly defined. Is the test discriminating two patient groups binned by survival time, if so, what is the survival time cutoff? Was the ROC curve done to pick the threshold used in figure 5 or as a classifier for patient stratification? If the ROC curve is a prototype of a clinical tool for patient stratification, to be statistically valid, there should be some level of internal validation to avoid performance bias. It is unclear if this was done.
ANSWER: The ROC curve analysis has been removed.
Comment 3. Mass spectrometry-based identification aptamer protein targets is not quantified so it is difficult to assess confidence of targets: Target proteins on CTCs were identified by CTC purification and lysis followed by capture with aptamers and identification by LC-MS/MS. They consider proteins to be specific targets of aptamers “if they are predominant in aptamer-derived samples compared to those obtained from controls (non-specific sequences and healthy controls)”. They indicate that CTCs from 7 LC patients (plus other controls) were analyzed in triplicate, but it is not indicated how the data were merged and analyzed and there are no quantitative data showing the level of enrichment. For example: Was plasma pooled across all 7 patients before mass spec or were each analyzed separately? Was the aptamer-specific enrichment identified in all three replicates of each patient (or in all three replicates of each patient pool)? How many different peptides per target were observed in each sample of each experiment? What was the log2 fold enrichment of aptamer versus control? Is there enough data to show significance of the enrichment with p-value or FDR? Without showing quantitative data, the data do not support the conclusion.
ANSWER: CTC samples from 7 patients were prepared and analyzed separately after incubation with the aptamers or the control sequence. The expression levels of the target proteins were not quantified across different patients due to the low number of isolated CTCs per patient and the limited sensitivity of the method. Proteins were selected as targets based on their presence in a larger number of experimental samples compared to the control samples. Blood samples from 4 healthy volunteers were used as an additional control; no aptamer-specific protein binding was detected in these controls.
Comment 4. Based on Table 5 the authors state that there is 50% identity between two proteins neutrophil defensin and peroxiredoxin; however, only 8 amino acids of each protein were compared. The full-length proteins are 30aa and 197aa, respectively. The data do not support the conclusion.
ANSWER: Thank you for this comment. You are correct that the protein sequence analysis requires a more comprehensive approach. Consequently, we have removed this data from the manuscript and plan to investigate this aspect in greater detail in future studies.
Minor points:
- Speed of centrifugation needs to be reported as x g and not rpm as rpm is specific to the centrifuge and not translatable to another lab. (e.g. Line 129 pg 3)
ANSWER: All rpm values in the text have been converted to their corresponding × g values.
- Images need magnification reported and size bars on the figures. In figure 1, are all panels with LC-17 and LC-18 labelling together, or are some panels labelled with LC-17 and some with LC-18?
ANSWER: As suggested, we have added a scale bar and caption to the figure. The staining was performed using both aptamers simultaneously.
- The data in table 2 describing identification of aptamer protein targets by MS from tumors has already been done (it is from a prior study) and thus should not be in results section.
ANSWER: As suggested, this data has been incorporated into the Discussion.
- Several times in the manuscript, it is stated that they have demonstrated that CTC count in the blood of LC patients correlates with the primary tumor spread, regional metastasis level, overall survival rates. Correlation is not the correct word because Mann Whitney u test does not measure correlation. For example, the authors could use the word dependence rather than correlation. An alternative is to use a different test (like spearman rank test for example) which would give a correlation measure. This would also likely yield a better p-value than the one they calculate using Mann Whitney U test (because by binning the data, power is lost).
ANSWER: Thank you for this suggestion. We have replaced the term "correlation" with "dependence" throughout the text.
- In Fig 6, AUC and p-value should be written on the figure or put in the legend.
ANSWER: The ROC curve analysis has been removed.
- There were no control subjects (i.e. those without LC) included in the experiment counting CTC cells per patient (Table 4). Zamay et al (ref 23) shows there are some CTCs detected in normal blood (~1.6 per 3 mL), which indicates that the CTCs detected in patients with low T or N stages could be false positives. It would have been good to include normal subjects to address this point.
ANSWER: Thank you for this feedback. We plan to conduct future studies using blood samples from healthy individuals and patients with other lung diseases. This point has been added to the Discussion section.
Round 2
Reviewer 1 Report
Comments and Suggestions for Authors
The author has fully and satisfactorily addressed all my previous review comments.